# Maternal Oxygen Administration during Labor: A Controversial Practice

**DOI:** 10.3390/children10081420

**Published:** 2023-08-21

**Authors:** Isabella Abati, Massimo Micaglio, Dario Giugni, Viola Seravalli, Giulia Vannucci, Mariarosaria Di Tommaso

**Affiliations:** 1Department of Health Sciences, Division of Obstetrics and Gynecology, Careggi Hospital, University of Florence, Largo Brambilla 3, 50134 Florence, Italy; isabella.abati@unifi.it (I.A.); viola.seravalli@unifi.it (V.S.); g.vannucci@unifi.it (G.V.); 2Department of Anesthesia and Intensive Care, Careggi Hospital, University of Florence, Largo Brambilla 3, 50134 Florence, Italy; massimo.micaglio@unifi.it (M.M.); giugnid@aou-careggi.toscana.it (D.G.)

**Keywords:** fetal heart rate, oxygen inhalation therapy, obstetric labor, placental circulation, umbilical cord blood

## Abstract

Oxygen administration to the mother is commonly performed during labor, especially in the case of a non-reassuring fetal heart rate, aiming to increase oxygen diffusion through the placenta to fetal tissues. The benefits and potential risks are controversial, especially when the mother is not hypoxemic. Its impact on placental gas exchange and the fetal acid–base equilibrium is not fully understood and it probably affects the sensible placental oxygen equilibrium causing a time-dependent vasoconstriction of umbilical and placental vessels. Hyperoxia might also cause the generation of radical oxygen species, raising concerns for the developing fetal cells. Moreover, this practice affects the maternal cardiovascular system, causing alterations of the cardiac index, heart rate and vascular resistance, and unclear effects on uterine blood flow. In conclusion, there is no evidence that maternal oxygen administration can provide any benefit in the case of a non-reassuring fetal heart rate pattern, while possible collateral effects warn of its utilization. Oxygen administration during labor should be reserved for cases of maternal hypoxia.

## 1. Introduction

Intrauterine resuscitation techniques comprise a series of interventions that are frequently used during labor. They are mainly adopted when fetal heart rate (FHR) is not reassuring, aiming to improve fetal well-being. In some cases, however, evidence of their benefits is weak; some of these measures have not been well-investigated and their use is grounded more on practical clinical knowledge than on evidence-based medicine [1].

Oxygen administration to the mother is one of such techniques, commonly performed in the aim of increasing oxygen diffusion through the placenta, from uterine circulation to fetal tissues.

Prophylactic oxygen administration has been proposed in many settings other than obstetrics, but it has often been abandoned due to poor evidence of its benefit and potential harm. In obstetrics, as well as for intrauterine resuscitation, it has been proposed to decrease surgical site infection in cesarean sections; however, the evidence is against this practice [2].

Moreover, the dose range, duration of administration and indications are variable, with no clear guideline. Nonetheless, this intervention is widely performed, with two out of three women in labor likely to receive oxygen at some point of labor in the United States [3]. Although information about other countries is scarce, many clinicians are inclined to use exogenous oxygen therapy in the case of category II FHR tracings or during the second stage of labor [3].

Most frequently, oxygen is administered at a dose of 10 L/min via a tight-fitting non-rebreather mask, ensuring a FiO_2_ concentration of 80–100% [4].

The benefits and potential risks of oxygen administration during active labor are controversial, especially when the mother is not hypoxemic [3,5]; indeed, evidence of clinical benefits is lacking, while concerns about its potential risks for the mother and fetus are growing. Clinical trials on its efficacy are hindered by the difficult identification of real oxygen tension in fetal and placental tissues.

This narrative review analyzes the effects of oxygen administration to the mother during labor, outlining the circumstances in which such administration is useful as an intrauterine resuscitation technique for the correction of a category II (indeterminate) FHR.

## 2. Intrauterine Resuscitation Techniques

During labor, fetal well-being is usually monitored via the interpretation of FHR patterns through cardiotocography (CTG), a non-invasive instrument for monitoring FHR and uterine contractile activity. FHR may be monitored intermittently in low-risk pregnancies when a one-to-one midwife–patient setting is guaranteed. In medium- and high-risk pregnancies, continuous monitoring of FHR during labor is recommended.

Most intrapartum CTG classifications are based on three classes of increasing fetal hypoxic risk. The first category refers to a physiological tracing with a high probability of a normal fetal acid–base status, while the third category consists of abnormal tracings that are highly predictive of a pathological fetal status, such as hypoxia or acidosis. Halfway, there is an intermediate category, that is referred to as “indeterminate”, “suspicious” or “category II FHR tracings” [1,6,7]. These tracings are very common in labor (>80% of women) and are considered to depict an intermediate risk of fetal acidemia. However, there is no consensus about their interpretation, and it is not clear in how many cases they really represent fetal distress [1].

When the condition of fetal hypoxia is likely but an emergency delivery is not indicated, a number of maneuvers and pharmacological treatments should be implemented in order to reduce potentially stressful intrauterine conditions. Straightforward reasoning should start from a diagnosis of the potential causes. Amongst those, maternal well-being should be investigated to exclude reduced oxygen delivery to the placenta, hypoglycemia, and potential signs and symptoms of chorioamnionitis. It is also useful to adopt a schematic approach, that should be simple and easy to remember. In case of the suspicion of a pathological fetal status, intrauterine resuscitation techniques are a group of simple and accessible interventions that aim to improve fetal oxygenation and well-being during labor. They include the interruption of oxytocin administration if this is in place and/or tocolysis with the purpose of reducing the frequency of uterine contractions, as well as maternal repositioning, exogenous oxygen administration and the use of intravenous fluids. Intrauterine resuscitation techniques are often used in case of the detection of a category II FHR tracing, and in selected cases of a category III FHR tracings, after the exclusion of acute events. Generally, they might be applied while continuing the course of labor, and they eventually revert FHR to a category I tracing. However, they might also be used to promote fetal well-being while accelerating delivery [8].

The pathophysiologic rationale is different for each intrauterine resuscitation technique [7], and their use might produce different effects. Maternal repositioning and intravenous fluids act on maternal circulation [9,10]. The interruption of oxytocin infusion and the administration of a tocolysis decreases uterine contractile activity [1]. The administration of exogenous oxygen to the mother is carried out in an aim to increase oxygen diffusion to the fetus, although great controversy surrounding this intervention exists. Notably, the efficacy of maternal oxygen administration, as well as of other intrauterine resuscitation techniques, has been evaluated in few randomized clinical trials, partly because it would not be reasonable to randomize a pregnant patient to receive no intervention during a category II o category III FHR tracing.

## 3. Fetal Oxygenation

Fetal oxygen delivery and uptake are the main drivers of fetal growth. The placenta is the center of physiologic exchange of oxygen, nutrients and metabolic waste between mother and fetus. Specifically, maternal blood is carried by the uterine arteries, which divide into 30–40 spiral arteries that deliver maternal blood into the intervillous space [11]. Blood traverses toward fetal villi and then percolates into a series of lobules, where exchanges of oxygen and nutrients with fetal circulation take place [11]. Fetal blood is carried by two umbilical arteries that divide into smaller vessels within the fetal villi. The human placenta is hemochorial, with three microscopic tissue layers, all originating from trophoblast cells, to separate maternal and fetal blood circulations. At the end of pregnancy, the villous surface increases to its maximum extent and the distance between maternal and fetal blood is as low as 4–5 micrometers in the terminal villi, to favor oxygen diffusion to the fetus.

Respiratory gases are transported via passive diffusion along the concentration gradient, with no energy requirement. Oxygen partial pressure in the fetal umbilical vein (UV) reflects diffusion from the maternal venous side of the spiral artery–capillary–vein loop [12]. Models of placental oxygen exchange revealed a great complexity; they have been largely based on sheep studies and, to a lesser extent, derived from human observational data or histological data on human placenta [12]. Different models exist. They differ in their assumptions of the actual effectiveness of the placenta in transporting oxygen to the fetus in vivo (the membrane oxygen diffusion capacity), and in their hypotheses of the direction of the blood stream. To simplify the latter concept, there are two possibilities: a “concurrent model” in which the fetal stream and maternal stream are in the same direction, or the opposite “countercurrent model” [12]. In the 1990s, the human placenta has was described as a “venous equilibrator” with a direct relationship between uterine venous and umbilical venous (UV) partial pressure of oxygen (PO_2_) [13]. The rate of exchange depends on several factors, either maternal, placental, or fetal, with two of the most important factors being uterine blood flow and maternal oxygen content [8].

The disruption of one of the factors can lead to a reduction in the rate of exchange, which eventually leads to fetal hypoxia and acidosis [8]. An important determinant of fetal oxygenation is maternal arterial PO_2_, which depends on adequate ventilation and pulmonary function. Disruptions of these factors are rare in obstetrics but may occur with maternal conditions such as anaphylactic shock, pulmonary disease, or heart failure. Severe maternal anemia may also represent a cause of reduced fetal oxygen availability. However, healthy women in labor have high blood oxygen saturation.

## 4. The Effect of Maternal Oxygen Administration on Fetal Acid-Base Status

The consequences of exogenous oxygen administration to the mother on fetal acid–base status have been debated for decades. Many data have been published, with a plethora of clinical variables, but with no agreement on the effects of administration of exogenous oxygen on fetal blood and tissues. Clinical studies on the relationship between oxygen supplementation and fetal acid–base studies are affected by the lack of standard methods and are mostly limited to observational non-randomized trials. A summary of the most relevant trials comparing fetal and neonatal outcomes of oxygen administration on fetal status during labor is presented in Table 1. 

Most authors focus on umbilical artery (UA) analysis at birth as an objective measure of fetal oxygenation and in utero metabolic status. Umbilical cord blood analysis has been proven helpful in improving the accuracy of fetal distress diagnosis and is closely related to neonatal prognosis. Hypoxemia, hypercapnia and acidosis are the main pathophysiological changes observed in neonatal asphyxia. This condition may lead to the occurrence of respiratory and circulatory disorders in newborns after delivery, which can cause a lack of spontaneous breathing or failure in the regular breathing pattern [14].

Early studies on oxygen administration and UA analysis (1967–1993) showed a beneficial effect on neonatal PO_2_ and pH, suggesting a benefit of hyperoxia on the fetus, especially when the fetus is already compromised [15,16,17,18,19,20]. In early 1967, a prospective study was conducted on pregnant women receiving 100% oxygen in term labor induced by oxytocin infusion. The study included both fetuses in normal conditions and fetuses with signs of fetal distress (based on their FHR patterns) and showed a positive effect of maternal hyperoxygenation in 10 cases of fetal tachycardia. In those cases, a significant but slight decrease in basal FHR was noted. Oxygen administration to such patients also reduced the frequency of late decelerations [15]. However, the authors also reported the occurrence of three cases of severely depressed newborns after very prolonged hyperoxygenation (>15 h) [15]. In six cases, the authors also measured fetal capillary pH and PO_2_ through the Saling technique of scalp sampling. They presented mixed results after oxygen administration: an increase in fetal pH in two cases, and a decrease in fetal pH and in the partial pressure of carbon dioxide (PCO_2_) in four cases. The fetal PO_2_ concentration increased in three cases but did not vary in two cases [15]. The limitations of the study are represented by the lack of a methodology and by the small sample of patients, that only included 21 pregnant women—of which only six cases were tested for fetal capillary pH and PO_2_ [15].

Khazin and colleagues also found a benefit of a few minutes of oxygen administration on late decelerations for FHR tracings during the second stage of labor in a study on 20 pregnant patients with oxygen administration. The authors also showed an increase in fetal PO_2_ without concomitant acidosis [16]. This study as well was limited by its observational nature, by the lack of methods (pH and P values were not reported) and by the small sample of patients [16].

In 1969, another study reported effects of exogenous oxygen administration on 103 laboring women (of which only 58 were healthy): 70 of these were given 100% oxygen and showed an overall increase in PO_2_ of 3.5 mmHg, obtained from fetal capillaries through the Saling technique of scalp sampling [17]. Interestingly, the study showed that the lower the initial fetal PO_2_, the greater the rise after oxygen administration [17]. The authors concluded that the hypoxic fetus allows an increased uptake of oxygen, while the well-oxygenated fetus has mechanisms which prevent its increased uptake [17]. However, it is important to consider that part of the women included in the analysis had various diseases, possibly constituting maternal hypoxemia (14 patients had toxemia, one had active tuberculosis, one had hypertension, and one had chronic renal disease) and the results were not presented separately based on maternal conditions. The hypoxic diseases of some patients affected the results because oxygen administration to such women raised the maternal arterial oxygen saturation to the normal status, while oxygen administration to healthy patients was expected to cause a hyperoxic state.

More recent studies have shown no effect on umbilical cord blood gas values. Two independent groups randomized delivering women to either receive oxygen at a low rate (2 L/min) versus sham supplementation or room air [21,22]. The first double-blinded trial was conducted on 443 patients and reported no significant difference in the UA pH between the two groups [21]. Moreover, oxygen supplementation did not affect FHR patterns [21]. The latter study was a randomized double-blinded trial on 56 patients and showed no significant difference in UA pH and UA PO_2_. However, the authors noticed a greater need for delivery room resuscitation in neonates born after maternal oxygen administration. Interestingly, they could not find a reason for this increased need for respiratory support [22]. Regarding the similar UA pH between the two groups, a possible explanation of the phenomenon is the mechanism of the placental equilibrium that might prevent fetal hyperoxia even when maternal blood is hyperoxic. However, how this mechanism works has not been defined yet.

In 2021, a systematic review and meta-analysis of 16 randomized clinical trials was published, comparing room air versus maternal oxygen supplementation in singleton, non-anomalous pregnant subjects undergoing delivery or elective cesarean section, comprising a total of 2052 patients [23]. The authors reported no significant difference in UA pH and in other neonatal outcomes [23]. However, they reported an increase in umbilical arterial oxygen pressure (UA PO_2_) in cases of elective cesarean section, which should not have any clinical benefit if not associated with a significant difference in UA pH. UA PO_2_ is a poor estimator of neonatal morbidity because hypoxia cannot be presumed from dissolved oxygen content alone. Indeed, it is only relevant when a decrease in UA PO_2_ ultimately leads to anaerobic metabolism and a decrease in UA pH [23]. The limitation of this systematic review is due to the great heterogeneity concerning the methods and outcomes between the included trials.

Differently, many authors reported that maternal oxygen therapy can worsen fetal acid–base status. Studies using human and animal models showed a decrease in neonatal pH and an increase in lactate levels in this situation. In 1995, Thorp and colleagues published a randomized controlled prospective trial on 86 women receiving face mask oxygen at 10 L/min or no oxygen at the onset of the second stage, showing an increase in neonatal acidemia in patients receiving oxygen. An UA pH of less than 7.20 was reported as being observed significantly more frequently in the oxygen group [24]. Likely, umbilical and placental vessels are extremely sensible and show hyperoxia-induced vasoconstriction, which might worsen placental oxygen exchange and fetal tissue perfusion. Indeed, as early as in 1963, Saling showed that maternal oxygen administration can cause a transient rise in PO_2_ followed by a drop in pH and a rise in the partial pressure of carbon dioxide PCO_2_ using fetal scalp sampling [25]. This might be explained by the development of placental vasoconstriction as a result of maternal hyperoxia [25].

**Table 1 children-10-01420-t001:** Summary of main trials comparing fetal and neonatal outcomes of oxygen administration.

Authors and Year of Publication	Number of Patients	Primary Outcome and Result (after Oxygen Administration)	Other Outcomes and Results (after Oxygen Administration)	Strengths	Limitations
Althabe et al., 1967 [15]	21 pregnancies with labor induction	Effect on fetal tachycardia and decelerations: slight decrease in fetal heart rate (FHR) and reduction in late decelerations Effect on fetal muscular partial pressure of oxygen (PO_2_) with polarographic method: rise in PO_2_	Effect on fetal capillary pH and PO_2_ (scalp sampling): mixed results (only in 6 patients)	Direct measurements of fetal PO_2_	Observational nature Lack of a control group Small number of patients
Gare et al., 1969 [17]	70 pregnancies in labor with oxygen administration Control group: 33 patients	Effect on fetal capillary pH and PO_2_ (scalp sampling): increase in 3.5 mmHg of PO_2_ after oxygen administration	Umbilical vein (UV) blood gas analysis: increase in PO_2_ after oxygen administration Umbilical artery (UA) blood gas analysis: no difference	Direct measurements of fetal PO_2_	Did not consider pulmonary or systemic disease separately No randomization
Thorp et al., 1995 [24]	Randomization of 86 pregnancies	Effect on UA pH: pH < 7.20 more frequent after oxygen administration	Effect on UA pH if administration lasted ≤10 min: increase in cord arterial pH Effect on UA pH if administration lasted > 10 min: decrease in cord arterial pH	Strong methodology	Small number of patients
Khaw et al., 2002 [26]	Double-blinded randomization of 44 patients undergoing cesarean section	Effect on reactive oxygen species activity in maternal blood, UA, and UV: increase in lipid hydroperoxides after oxygen administration	Effect of oxygen on maternal arterial PO_2_: higher after oxygen administration	Strong methodology	Small number of patients Clinical relevance of the finding needs to be assessed
Nesterenko et al., 2012 [22]	Double-blinded randomization of 56 pregnancies in labor	Effect on maternal and umbilical antioxidant enzymes concentration: no difference Effect on UA pH and PO_2_: no difference	Effect on need for newborn resuscitation: higher need after oxygen administration	Strong methodology	Small number of patients
Qian et al., 2017 [21]	Double-blinded randomization of 443 pregnancies in labor	Effect on UA pH: no difference	Effect on FHR patterns: no difference	Strong methodology High number of patients	
Raghuraman et al., 2018 [27] Watkins et al., 2020 (secondary analysis) [28]	Randomization of 99 pregnancies developing type II FHR tracings during labor	Effect on UA lactate: no difference	Effect on other UA blood gas analysis: no difference Effect on mode of delivery: no difference Effect on UA and UV PO_2_ if administration lasted for a long period: lower UV PO_2_ after long oxygen administration	Accurate selection of cases with type II FHR Exclusion of cases with maternal hypoxia	Unblinded trial
Moors et al., 2020 [29]	Randomization of 117 pregnancies developing type II and type III FHR tracings	Effect on FHR pattern: significant improvement on FHR tracings	Effect on Apgar score: no difference Effect on UA and UV pH, base excess and PCO_2_: no difference	Accurate selection of type II and type III FHR tracings	Unblinded trial Only addressed second stage of labor

## 5. Oxygen Therapy in Non-Reassuring CTG

In clinical practice, oxygen therapy is often used as an intrauterine resuscitation measure during delivery, aiming to improve fetal oxygenation when there is a concern about fetal status. Clinicians often administer oxygen in the presence of a non-reassuring FHR pattern. Many of the previously discussed studies were limited by the fact that they did not specifically consider abnormal CTG tracings, and hypotheses exist of a possible therapeutic effect of oxygen in the case of alterations of fetoplacental gas exchanges.

During the period 2016–2018, a randomized trial on 117 singleton pregnancies in a single European center reported the effects of oxygen administration in the case of type II or type III FHR tracings during the second stage of labor [29]. The authors noted a positive effect on FHR tracings and the presence of fewer episiotomies in the oxygen group [29]. FHR tracings were accurately analyzed by three blinded clinicians, until a consensus was reached; however, the authors stated that FHR improvements did not necessarily correlate to a better fetal metabolic status [29]. In fact, there was no significant difference in UA gas components and the newborn and Apgar score in the two different groups, suggesting that exogenous oxygen did not improve fetal status [29].

In 2016–2017, Raghuraman and colleagues conducted a randomized, unblinded non-inferiority clinical trial in a single tertiary health care center including 99 patients with singleton, term pregnancy developing type II FHR tracings during labor [27].

Patients were randomized in a ratio of 1:1 to breathing room air (51 women) or oxygen (48 women) [27]. The primary outcome was the measure of UA lactate, considered the most valuable index of metabolic acidosis (since it changes earlier than pH does) and neonatal hypoxia-associated morbidity. The results showed similar lactate values in both groups (30.6 mg/dl in oxygen vs. 31.5 mg/dl in room air) [27]. Other UA gas components were also similar in the two groups, as well as were the modes of delivery [27]. This trial suggested that oxygen supplementation is not useful even in the setting of abnormal FHR tracings [27]. The authors hypothesized that the hemoglobin dissociation curve might explain the inefficacy of oxygen therapy [27]. If maternal oxygenation is normal, extra oxygen only causes a minimal increase in maternal arterial oxygen pressure, as well as in fetal oxygen pressure. The strengths of this study are the accurate selection of category II FHR tracings and the exclusion of pregnancies affected by maternal hypoxia.

A planned secondary analysis of this study also showed that oxygen administration was not associated with an improvement of high-risk category II FHR features (including recurrent variable or late decelerations, prolonged decelerations, tachycardia or minimal variability) or the resolution of recurrent decelerations within 60 min in electronic fetal monitoring patterns [30]. Indeed, most cases with recurrent decelerations were resolved without exogenous oxygen administration [30].

## 6. Timing of Oxygen Administration

To further complicate the issue of oxygen therapy, studies showed that its effect might vary in a time-dependent fashion. In as early as 1995, Thorp and colleagues showed a deterioration in cord blood values after oxygen administration for >10 min, which was not present for shorter periods; this led the authors to suggest that uteroplacental or fetoplacental vasoconstriction might take some time to occur [24].

A planned secondary analysis of the previously described trial by Raghuraman and colleagues was also conducted to investigate cord blood oxygenation and acid–base values after different durations of O_2_ exposure (cutoff: 75th percentile, 176 min) [28]. The authors concluded that newborns with longer durations of O_2_ exposure had a lower partial pressure of O_2_ in the UV (25.5 mmHg in the 12 patients receiving oxygen for >176 min vs. 32.5 mmHg in the 36 patients receiving oxygen for <176 min) [28]. This finding again suggested that impaired placental O_2_ transfer occurs after prolonged O_2_ exposure [28].

## 7. Generation of Reactive Oxygen Species

The potential risks of maternal oxygen administration also include the generation of reactive oxygen species (ROS) that can eventually lead to structural cell damages [4]. The administration of oxygen to the mother causes maternal hyperoxia, and this might increase free radical activity both in the mother and the fetus.

Because of an intrinsic lack of antioxidant enzymes, fetal and neonatal neurons are particularly sensitive to oxidative stress. Stress can arise from a number of different pathways [4]. Hypoxia-induced and consequent ischemia–reperfusion damage leads to ROS formation through a classical pathway, with the generation of purine metabolites. Otherwise, hyperoxia leads to an alternative pathway with no purine metabolites [26]. It is hard to directly measure ROS in blood samples, but it is possible to detect the products of their attack, such as lipid hydroperoxides.

Khaw and colleagues conducted a double-blinded trial with the analysis of ROS activity in both neonatal and maternal blood samples of patients undergoing elective cesarean section, that were randomized to breathe room air or FiO_2_ at a concentration of 60%. In the oxygen group, they detected an increase in lipid hydroperoxides (malondialdehyde, isoprostane, and organic hydroperoxides) that was more pronounced in umbilical blood than in maternal blood, suggesting that generation occurred in the fetoplacental unit. Given that the highest concentration was found in the UV, they suggested that ROS production and activity was greatest in the placenta, which represents the interface with highest hyperoxic distress. Since they did not detect any purine metabolite activity, the authors also excluded a concomitant pathogenic pathway involving hypoxia [26].

This important study suggests that breathing high concentrations of FiO_2_ can damage both fetal and maternal cell structures via hyperoxic oxidative stress. This is in line with previous concerns about neonatal oxygen resuscitation that can lead to adverse effects of hyperoxia [31,32].

Further information came from animal models. In 2003, a study conducted on goats analyzed lipid peroxidation in the case of maternal hyperoxygenation in the presence of fetal asphyxia (caused via cord occlusion). The authors showed an increase in lipid peroxidation with exogenous oxygen administration [33]. A sheep model of fetal asphyxia obtained via umbilical cord occlusion analyzed adenosine and hypoxanthine production from adenosine 5′-triphosphate (ATP) [34]. Hypoxanthine, a source of ROS, was higher in ewes with maternal oxygen administration during the period of recovery from asphyxia [35]. The authors concluded that oxygen therapy may be inadequate for the management of fetuses with variable decelerations [34].

On the other hand, it is expectable that when oxygen administration is provided for maternal hypoxia at the right dosage, it does not cause a rise in ROS activity, because it does not induce hyperoxia.

## 8. Cardiovascular and Respiratory Effects of Maternal Hyperoxygenation

During normal pregnancy, major hemodynamic changes occur, such as an increase in blood volume, cardiac output and heart rate, as well as do reductions in systemic vascular resistance and blood pressure [35]. Furthermore, labor and delivery are associated with further significant modifications due to emotional distress and uterine activity, such as a greater rise in cardiac output and heart rate [35]. Regarding the respiratory system, the rate of respiration is similar to that in the pre-pregnancy state, but a decrease in arterial carbon dioxide tension can be noted [35]. These physiological changes are crucial for the extensive metabolic demands of pregnancy [35].

The administration of maternal oxygen during the third trimester has been linked to several different changes of the cardiovascular system and to variations in respiration.

In 2005, Simchen and colleagues evaluated the respiratory results of brief hyperoxygenation on eight healthy near-term pregnant women and they reported resulting hypocapnia and hyperventilation [36]. In 2019, Mchugh and colleagues showed that hyperoxygenation during the third trimester causes not only a decrease in the maternal heart rate and cardiac index, but also a rise in systemic vascular resistance, both without recovery after the cessation of oxygen administration [5]. They performed noninvasive hemodynamic monitoring during a brief period of hyperoxygenation in 46 women with singleton, non-anomalous, third-trimester pregnancies and compared them to 20 nonpregnant controls [5]. The authors hypothesized that lower heart rate is probably caused by increased vagal activity, while increased systemic vascular resistance depends either on the generation of ROS or on the long-lasting calcium channels that cause vasoconstriction [5].

These demonstrated cardiovascular effects warn against maternal oxygen administration because of the potential maternal complications, especially if it is given without proper hemodynamic monitoring. Furthermore, the effects on uterine blood flow still need further studies because they have not been assessed yet.

A summary of the different effects of oxygen administration to patients with normal saturation can be found in Figure 1.

## 9. Conclusions

There is no evidence that maternal oxygen administration can have any benefit in case of a non-reassuring FHR pattern. In recent decades, many investigators studied the impact of oxygen administration to the mother on fetal status during labor. However, trials showed conflicting as well as worrying results. Furthermore, other possible collateral effects on the newborn and on the mother warn against its usage. Recent neonatal and maternal studies showed that hyperoxygenation might be detrimental to both types of patients, because it can cause ROS generation which eventually leads to structural cell damage, as well as potential consequences on maternal cardiovascular and respiratory systems.

To conclude, maternal oxygen administration should be reserved for cases of maternal hypoxia, as suggested by many authors and guidelines [1,3,5,6,37,38], with the aim to restore normal oxygen saturation, and therefore ensure oxygen delivery to the placenta and the fetus.

Given the potential harmful effects of hyperoxygenation, especially if oxygen is administrated for a long time, we suggest its use in labor only in selected cases when the cause of the type II FHR tracings is due to fetal hypoxia secondary to maternal hypoxia.

## 10. Future Directions

In the future, it is advisable that all guidelines will discourage the use of maternal oxygen administration to normoxic patients in labor, while favoring other techniques in the case of a type II FHR tracing. Regarding other intrauterine resuscitation techniques, such as tocolysis, maternal repositioning and intravenous fluids, their positive influence has already been established [1,9,10]. However, proper clinical trials could improve their applications; it would be helpful, for instance, to assess the best type, quantity, and rate of fluids to use, as well as the most appropriate tocolytic agent for different clinical situations.

On the other hand, the clinician must administer exogenous oxygen in cases of maternal hypoxia, aiming to restore a normal oxygen saturation. Therefore, it would be interesting to study the fastest and most effective way to correct the hypoxia without causing any detrimental consequences to the mother and the fetus. Research should focus on different variables of interest, such as baseline maternal saturation and the duration of administration. The hemodynamic effects on different maternal vascular districts, particularly on the uterine circulation, should be further investigated, to ensure that exogenous oxygen does not bear any negative consequences. In this context, observational studies may have a significant role. However, future randomized clinical trials should also be designed, with the purpose of studying the effect of this intervention on short- and long-term clinical outcomes, and of excluding potential harmful effects on the fetus.

## Figures and Tables

**Figure 1 children-10-01420-f001:**
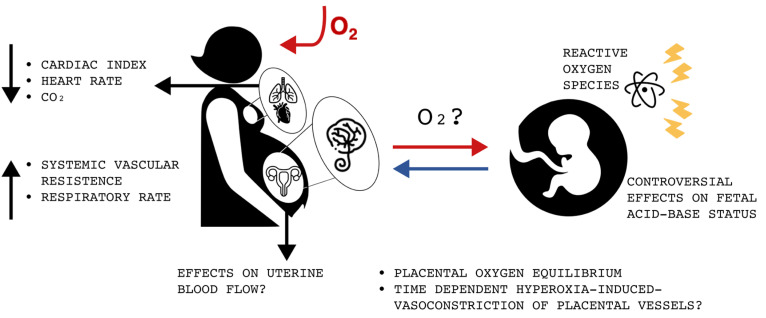
Effects of exogenous oxygen administration to normoxic pregnant women during labor (O_2_: oxygen; CO_2_: carbon dioxide).

## Data Availability

No new data were created or analyzed in this study. Data sharing is not applicable to this article.

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
