# Peer review of "Maternal Oxygen Administration during Labor: A Controversial Practice"

_children, 2023, doi:10.3390/children10081420_

Round 1
Reviewer 1 Report
Review report for autors:
A brief summary:
The aim of this study is to provide an overview of available literature reserch on the topic of oxygen administration to mothers during labor and its effects on the fetus. The conclusion of the autor of this rewiew is that there is no evidence that maternal oxygen administration can provide any benefit in case of a non-reassuring fetal heart rate pattern, while possible collateral effects warn on its utilization. Oxygen administration during labor should be reserved for cases of maternal hypoxia.
This rewiew paper is a very detailed and schematically structurated presentation of all available reserch relevant to this topic. This topic has been very little researched until now, due to the limitations that the autors themselfs stated in the paper, with questionable moral principes of futher research. This issue is extremely interesting and the conclusion is applicable in the work of all delivery room staff who are participants in childbirth.
General concept comments:
The researchers clearly stated the limitations of the study which are self-evident, from the ethical and moral component. Directions for future research that can be conducted are clearly highlighted.
The title is well written, it is concise.
The summary is well writen and contains all the important and most significant and relevant literature dana obtained by searching and comparing the literature, with a clear conclusion based on the same.
The introduction is clearly and interestingly written, with minor additions it provides an appropriate overview and introduction to the main topic of the paper.
In line 54., 55., in my opinion, it would be interesting to know if there are similar data and resarch conducted in the researchers country or in some of the EU countries.
This is interesting rewiew paper, very well structured, schematically well supported, explained in detail on a theoretical and clinically applicable level.
In line 103.,104. the reason why such reasearch is not carried out is clearly emphasized, the only question is wheather there are available research carried out in the EU?
The tables are systematically and clearly written, only they are not written chronologically.
In this resarch is an excellent schematic representation (figure) that looks back on the previously discussed patophysiological aspect of the effect of oxygen on the mother and child.
In the conclusion itself, it is clearly emphasized in the conclusion in which situations oxygen is a therapeutic then it is used, and what the dangers of its use have been indicated so far.
The literature is correctly cited. Slightly more than 27% of references (10/37) are references of recent publications. I believe that references should be more included in the paper, as well as data from similar research carried out in EU countries.
The conclusion is consistent, consize with clearly presented arguments.
The hypothesis of the article is clearly written (the aim of the study), explained and later analyzed by comparing different researches.
Experimental design of the study is appropriate for this type of reasearch (narrative rewiew).
This study is a new way of seeing this clinicaly procedure. Applicable to a smaller number of specialists, especially perinatologysts and other stuff working in the delivery rooms.
In conclusion, I belive that this is an interesting work, with a new perspective in the views of clinicians applicable in a narrower, strictly defined population, and I recommend its acceptance for publications after minor corrections.
Reviewer 2 Report
I want to thank the Editor for the opportunity to review the article titled "Maternal Oxygen Administration During Labor: A Controversial Practice".
Moreover, I congratulate the authors for writing a detailed and well-organized narrative review on a significant yet controversial topic in obstetrical practice. As clinicians, we encounter indeterminate CTGs almost every day and inevitably need to perform some of the mentioned intrauterine resuscitation techniques. This narrative review could be helpful for OBGYN interns and young specialists since oxygen administration is the first and, in some cases, the only resuscitation technique available.
The manuscript's introduction provides enough background information for the review. All manuscript sections are well-written and logically organized. English writing is fine, and there are no major grammatical issues detected.
The only minor proposal to fully cover the manuscript's topic would be the addition of a short paragraph on maternal respiratory and cardiovascular changes observed during normal pregnancy.
